# Sample Complexity of Asynchronous Q-Learning: Sharper Analysis and Variance Reduction

**Gen Li**
Tsinghua

**Yuting Wei**
CMU

**Yuejie Chi**
CMU

**Yuantao Gu**
Tsinghua

**Yuxin Chen**
Princeton

## Abstract

Asynchronous Q-learning aims to learn the optimal action-value function (or Q-function) of a Markov decision process (MDP), based on a single trajectory of Markovian samples induced by a behavior policy. Focusing on a $\gamma$-discounted MDP with state space $\mathcal{S}$ and action space $\mathcal{A}$, we demonstrate that the $\ell_\infty$-based sample complexity of classical asynchronous Q-learning — namely, the number of samples needed to yield an entrywise $\varepsilon$-accurate estimate of the Q-function — is at most on the order of

$$\frac{1}{\mu_{\mathsf{min}}(1-\gamma)^5\varepsilon^2} + \frac{t_{\mathsf{mix}}}{\mu_{\mathsf{min}}(1-\gamma)}$$

up to some logarithmic factor, provided that a proper constant learning rate is adopted. Here, $t_{\mathsf{mix}}$ and $\mu_{\mathsf{min}}$ denote respectively the mixing time and the minimum state-action occupancy probability of the sample trajectory. The first term of this bound matches the complexity in the case with independent samples drawn from the stationary distribution of the trajectory. The second term reflects the expense taken for the empirical distribution of the Markovian trajectory to reach a steady state, which is incurred at the very beginning and becomes amortized as the algorithm runs. Encouragingly, the above bound improves upon the state-of-the-art result by a factor of at least $|\mathcal{S}||\mathcal{A}|$. Further, the scaling on the discount complexity can be improved by means of variance reduction.

## 1 Introduction

Model-free algorithms such as Q-learning [46] play a central role in recent breakthroughs of reinforcement learning (RL) [32]. In contrast to model-based algorithms that decouple model estimation and planning, model-free algorithms attempt to directly interact with the environment — in the form of a policy that selects actions based on perceived states of the environment — from the collected data samples, without modeling the environment explicitly. Therefore, model-free algorithms are able to process data in an online fashion and are often memory-efficient. Understanding and improving the sample efficiency of model-free algorithms lie at the core of recent research activity [19], whose importance is particularly evident for the class of RL applications in which data collection is costly and time-consuming (such as clinical trials, online advertisements, and so on).

This paper concentrates on Q-learning — an off-policy model-free algorithm that seeks to learn the optimal action-value function by observing what happens under a behavior policy. The off-policy feature makes it appealing in various RL applications where it is infeasible to change the policy under evaluation on the fly. There are two basic update models in Q-learning. The first one is termed a *synchronous* setting, which hypothesizes on the existence of a simulator (or a generative model); at each time, the simulator generates an *independent* sample for every state-action pair, and the estimates are updated simultaneously across all state-action pairs. The second model concerns an *asynchronous* setting, where only a single sample trajectory following a behavior policy is accessible; at each time, the algorithm updates its estimate of a single state-action pair using one state transition from the

| Paper | Sample complexity | Learning rate |
|---|---|---|
| Even-Dar and Mansour (2003) [20] | $\frac{(t_{\text{cover}})^{\frac{1}{1-\gamma}}}{(1-\gamma)^4\varepsilon^2}$ | linear: $\frac{1}{t}$ |
| Even-Dar and Mansour (2003) [20] | $\left(\frac{t_{\text{cover}}^{1+3\omega}}{(1-\gamma)^4\varepsilon^2}\right)^{\frac{1}{\omega}} + \left(\frac{t_{\text{cover}}}{1-\gamma}\right)^{\frac{1}{1-\omega}}$ | polynomial: $\frac{1}{t^\omega}, \omega \in (\frac{1}{2}, 1)$ |
| Beck and Srikant (2012) [5] | $\frac{t_{\text{cover}}^3 \lvert\mathcal{S}\rvert\lvert\mathcal{A}\rvert}{(1-\gamma)^5\varepsilon^2}$ | constant: $\frac{(1-\gamma)^4\varepsilon^2}{\lvert\mathcal{S}\rvert\lvert\mathcal{A}\rvert t_{\text{cover}}^2}$ |
| Qu and Wierman (2020) [35] | $\frac{t_{\text{mix}}}{\mu_{\text{min}}^2(1-\gamma)^5\varepsilon^2}$ | rescaled linear: $\frac{\frac{1}{\mu_{\text{min}}(1-\gamma)}}{t+\max\{\frac{1}{\mu_{\text{min}}(1-\gamma)}, t_{\text{mix}}\}}$ |
| **This work (Theorem 1)** | $\frac{1}{\mu_{\text{min}}(1-\gamma)^5\varepsilon^2} + \frac{t_{\text{mix}}}{\mu_{\text{min}}(1-\gamma)}$ | constant: $\min\left\{\frac{(1-\gamma)^4\varepsilon^2}{\gamma^2}, \frac{1}{t_{\text{mix}}}\right\}$ |
| **This work (Theorem 2)** | $\frac{t_{\text{cover}}}{(1-\gamma)^5\varepsilon^2}$ | constant: $\min\left\{\frac{(1-\gamma)^4\varepsilon^2}{\gamma^2}, 1\right\}$ |

Table 1: Sample complexity of asynchronous Q-learning to compute an $\varepsilon$-optimal Q-function in the $\ell_\infty$ norm, where we hide all logarithmic factors. With regards to the Markovian trajectory induced by the behavior policy, we denote by $t_{\text{cover}}$, $t_{\text{mix}}$, and $\mu_{\text{min}}$ the cover time, mixing time, and minimum state-action occupancy probability of the associated stationary distribution, respectively.

trajectory. Obviously, understanding the asynchronous setting is considerably more challenging than the synchronous model, due to the *Markovian* (and hence non-i.i.d.) nature of its sampling process.

Focusing on an infinite-horizon Markov decision process (MDP) with state space $\mathcal{S}$ and action space $\mathcal{A}$, this work investigates asynchronous Q-learning on a single *Markovian trajectory*. We ask a fundamental question:

*How many samples are needed for asynchronous Q-learning to learn the optimal Q-function?*

Despite a considerable amount of prior work exploring this algorithm (ranging from the classical work [24, 43] to the very recent paper [35]), it remains unclear whether existing sample complexity analysis of asynchronous Q-learning is tight. As we shall elucidate momentarily, there exists a large gap — at least as large as $\lvert\mathcal{S}\rvert\lvert\mathcal{A}\rvert$ — between the state-of-the-art sample complexity bound for asynchronous Q-learning [35] and the one derived for the synchronous counterpart [44]. This raises a natural desire to examine whether there is any bottleneck intrinsic to the asynchronous setting that significantly limits its performance.

**Our contributions.** This paper develops a refined analysis framework that sharpens our understanding about the sample efficiency of classical asynchronous Q-learning on a single sample trajectory. Setting the stage, consider an infinite-horizon MDP with state space $\mathcal{S}$, action space $\mathcal{A}$, and a discount factor $\gamma \in (0, 1)$. What we have access to is a sample trajectory of the MDP induced by a stationary behavior policy. In contrast to the synchronous setting with i.i.d. samples, we single out two parameters intrinsic to the Markovian sample trajectory: *(i)* the mixing time $t_{\text{mix}}$, which characterizes how fast the trajectory disentangle itself from the initial state; *(ii)* the smallest state-action occupancy probability $\mu_{\text{min}}$ of the stationary distribution of the trajectory, which captures how frequent each state-action pair has been at least visited.

With these parameters in place, our findings unveil that: the sample complexity required for asynchronous Q-learning to yield an $\varepsilon$-optimal Q-function estimate – in a strong $\ell_\infty$ sense – is at most[1]

$$\widetilde{O}\left(\frac{1}{\mu_{\text{min}}(1-\gamma)^5\varepsilon^2} + \frac{t_{\text{mix}}}{\mu_{\text{min}}(1-\gamma)}\right). \tag{1}$$

The first component of (1) is consistent with the sample complexity derived for the setting with independent samples drawn from the stationary distribution of the trajectory [44]. In comparison, the second term of (1) — which is unaffected by the accuracy level $\varepsilon$ — is intrinsic to the Markovian nature of the trajectory; in essence, this term reflects the cost taken for the empirical distribution of the sample trajectory to converge to a steady state, and becomes amortized as the algorithm runs. In other words, the behavior of asynchronous Q-learning would resemble what happens in the setting with independent samples, as long as the algorithm has been run for reasonably long.

Furthermore, we leverage the idea of variance reduction to improve the scaling with the discount complexity $\frac{1}{1-\gamma}$. We demonstrate that a variance-reduced variant of asynchronous Q-learning attains $\varepsilon$-accuracy using at most

$$\widetilde{O}\Big( \frac{1}{\mu_{\mathsf{min}}(1-\gamma)^3 \min\{1,\varepsilon^2\}} + \frac{t_{\mathsf{mix}}}{\mu_{\mathsf{min}}(1-\gamma)} \Big) \qquad (2)$$

samples, matching the complexity of its synchronous counterpart if $\varepsilon \leq \min\big\{1, \frac{1}{(1-\gamma)\sqrt{t_{\mathsf{mix}}}}\big\}$ [45]. Moreover, by taking the action space to be a singleton set, the above results immediately lead to $\ell_\infty$-based sample complexity for temporal difference (TD) learning [41] on Markovian samples; the interested reader is referred to [**?** ] for more details.

Due to the space limits, the proofs of all theorems are deferred to the full version [**?** ].

**Comparisons with past work.** A large fraction of the classical literature focused on asymptotic convergence analysis of asynchronous Q-learning (e.g. [24, 42, 43]); these results, however, did not lead to non-asymptotic sample complexity bounds. The state-of-the-art sample complexity analysis was due to the recent work [35], which derived a sample complexity bound $\widetilde{O}\big(\frac{t_{\mathsf{mix}}}{\mu_{\mathsf{min}}^2(1-\gamma)^5\varepsilon^2}\big)$. Given the obvious lower bound $1/\mu_{\mathsf{min}} \geq |\mathcal{S}||\mathcal{A}|$, our result (1) improves upon that of [35] by a factor at least on the order of $|\mathcal{S}||\mathcal{A}| \min\big\{t_{\mathsf{mix}}, \frac{1}{(1-\gamma)^4\varepsilon^2}\big\}$. In addition, we note that several prior work [5, 20] developed sample complexity bounds in terms of the cover time $t_{\mathsf{cover}}$ of the sample trajectory, namely, the time taken for the trajectory to visit all state-action pairs at least once. Our analysis framework readily yields another sample complexity bound $\widetilde{O}\big(\frac{t_{\mathsf{cover}}}{(1-\gamma)^5\varepsilon^2}\big)$, which strengthens the existing bounds by a factor of at least $t_{\mathsf{cover}}^2|\mathcal{S}||\mathcal{A}| \geq |\mathcal{S}|^3|\mathcal{A}|^3$. See Table 1 for detailed comparisons.

## 2 Models and background

This paper studies an infinite-horizon MDP with discounted rewards, as represented by a quintuple $\mathcal{M} = (\mathcal{S}, \mathcal{A}, P, r, \gamma)$. Here, $\mathcal{S}$ and $\mathcal{A}$ denote respectively the (finite) state space and action space, whereas $\gamma \in (0,1)$ indicates the discount factor. We use $P : \mathcal{S} \times \mathcal{A} \to \Delta(\mathcal{S})$ to represent the probability transition kernel of the MDP, where for each state-action pair $(s,a) \in \mathcal{S} \times \mathcal{A}$, $P(s' \mid s, a)$ denotes the probability of transiting to state $s'$ from state $s$ when action $a$ is executed. The reward function is represented by $r : \mathcal{S} \times \mathcal{A} \to [0,1]$, such that $r(s,a)$ denotes the immediate reward from state $s$ when action $a$ is taken; for simplicity, we assume throughout that all rewards lie within $[0,1]$. We focus on the tabular setting which, despite its basic form, is not yet well understood.

**Q-function and the Bellman operator.** An action selection rule is termed a *policy* and represented by a mapping $\pi : \mathcal{S} \to \Delta(\mathcal{A})$, which maps a state to a distribution over the set of actions. A policy is said to be *stationary* if it is time-invariant. We denote by $\{s_t, a_t, r_t\}_{t=0}^\infty$ a sample trajectory, where $s_t$ (resp. $a_t$) denotes the state (resp. the action taken), and $r_t = r(s_t, a_t)$ denotes the reward received at time $t$. It is assumed throughout that the rewards are deterministic and depend solely upon the current state-action pair. We denote by $V^\pi : \mathcal{S} \to \mathbb{R}$ the value function of a policy $\pi$, namely,

$$\forall s \in \mathcal{S} : \qquad V^\pi(s) := \mathbb{E}\left[ \sum_{t=0}^\infty \gamma^t r(s_t, a_t) \,\big|\, s_0 = s \right],$$

which is the expected discounted cumulative reward received when *(i)* the initial state is $s_0 = s$, *(ii)* the actions are taken based on the policy $\pi$ (namely, $a_t \sim \pi(s_t)$ for all $t \geq 0$) and the trajectory is generated based on the transition kernel (namely, $s_{t+1} \sim P(\cdot|s_t, a_t)$). It can be easily verified that $0 \leq V^\pi(s) \leq \frac{1}{1-\gamma}$ for any $\pi$. The action-value function (also Q-function) $Q^\pi : \mathcal{S} \times \mathcal{A} \to \mathbb{R}$ of a policy $\pi$ is defined by

$$\forall (s,a) \in \mathcal{S} \times \mathcal{A} : \qquad Q^\pi(s,a) := \mathbb{E}\left[ \sum_{t=0}^\infty \gamma^t r(s_t, a_t) \,\big|\, s_0 = s, a_0 = a \right],$$

where the actions are taken according to the policy $\pi$ except the initial action (i.e. $a_t \sim \pi(s_t)$ for all $t \geq 1$). As is well-known, there exists an optimal policy — denoted by $\pi^\star$ — that simultaneously maximizes $V^\pi(s)$ and $Q^\pi(s,a)$ uniformly over all state-action pairs $(s,a) \in (\mathcal{S} \times \mathcal{A})$. Here and throughout, we shall denote by $V^\star := V^{\pi^\star}$ and $Q^\star := Q^{\pi^\star}$ the optimal value function and the

optimal Q-function, respectively. In addition, the Bellman operator $\mathcal{T}$, which is a mapping from $\mathbb{R}^{|\mathcal{S}| \times |\mathcal{A}|}$ to itself, is defined such that the $(s, a)$-th entry of $\mathcal{T}(Q)$ is given by

$$\mathcal{T}(Q)(s, a) := r(s, a) + \gamma \mathbb{E}_{s' \sim P(\cdot | s, a)} \Big[ \max_{a' \in \mathcal{A}} Q(s', a') \Big].$$

It is well known that the optimal Q-function $Q^\star$ is the unique fixed point of the Bellman operator.

**Sample trajectory and behavior policy.** Imagine we have access to a sample trajectory $\{s_t, a_t, r_t\}_{t=0}^\infty$ generated by the MDP $\mathcal{M}$ under a given stationary policy $\pi_{\mathsf{b}}$ — called a *behavior policy*. The behavior policy is deployed to help one learn the "behavior" of the MDP under consideration, which often differs from the optimal policy being sought. Given the stationarity of $\pi_{\mathsf{b}}$, the sample trajectory can be viewed as a sample path of a time-homogeneous Markov chain over all state-action pairs. Throughout this paper, we impose the following assumption [34].

**Assumption 1.** *The Markov chain induced by the stationary behavior policy $\pi_{\mathsf{b}}$ is uniformly ergodic.*

There are several properties concerning the behavior policy and its resulting Markov chain that play a crucial role in learning the optimal Q-function. Specifically, denote by $\mu_{\pi_{\mathsf{b}}}$ the stationary distribution (over all state-action pairs) of the aforementioned behavior Markov chain, and define

$$\mu_{\mathsf{min}} := \min_{(s, a) \in \mathcal{S} \times \mathcal{A}} \mu_{\pi_{\mathsf{b}}}(s, a). \tag{3}$$

Intuitively, $\mu_{\mathsf{min}}$ reflects an information bottleneck — the smaller $\mu_{\mathsf{min}}$ is, the more samples are needed in order to ensure all state-action pairs are visited sufficiently many times. In addition, we define the associated mixing time of the chain as

$$t_{\mathsf{mix}} := \min \Big\{ t \ \Big| \ \max_{(s_0, a_0) \in \mathcal{S} \times \mathcal{A}} d_{\mathsf{TV}} \big( P^t(\cdot | s_0, a_0), \mu_{\pi_{\mathsf{b}}} \big) \leq \frac{1}{4} \Big\}, \tag{4}$$

where $P^t(\cdot | s_0, a_0)$ denotes the distribution of $(s_t, a_t)$ conditional on the initial state-action pair $(s_0, a_0)$, and $d_{\mathsf{TV}}(\mu, \nu)$ stands for the total variation distance between two distributions $\mu$ and $\nu$ [34]. In words, the mixing time $t_{\mathsf{mix}}$ captures how fast the sample trajectory decorrelates from its initial state. Moreover, we define the cover time associated with this Markov chain as follows

$$t_{\mathsf{cover}} := \min \Big\{ t \ \Big| \ \min_{(s_0, a_0) \in \mathcal{S} \times \mathcal{A}} \mathbb{P} \big( \mathcal{B}_t | s_0, a_0 \big) \geq \frac{1}{2} \Big\}, \tag{5}$$

where $\mathcal{B}_t$ denotes the event such that all $(s, a) \in \mathcal{S} \times \mathcal{A}$ have been visited at least once between time 0 and time $t$, and $\mathbb{P}\big(\mathcal{B}_t | s_0, a_0\big)$ denotes the probability of $\mathcal{B}_t$ conditional on the initial state $(s_0, a_0)$.

**Goal.** Given *a single* sample trajectory $\{s_t, a_t, r_t\}_{t=0}^\infty$ generated by the behavior policy $\pi_{\mathsf{b}}$, we aim to compute/approximate the optimal Q-function $Q^\star$ in an $\ell_\infty$ sense. The current paper focuses on characterizing, in a non-asymptotic manner, the sample efficiency of classical Q-learning and its variance-reduced variant.

## 3  Asynchronous Q-learning on a single trajectory

**Algorithm.** The Q-learning algorithm [46] is arguably one of the most famous off-policy algorithms aimed at learning the optimal Q-function. Given the Markovian trajectory $\{s_t, a_t, r_t\}_{t=0}^\infty$ generated by the behavior policy $\pi_{\mathsf{b}}$, the asynchronous Q-learning algorithm maintains a Q-function estimate $Q_t : \mathcal{S} \times \mathcal{A} \to \mathbb{R}$ at each time $t$ and adopts the following iterative update rule

$$\begin{aligned} Q_t(s_{t-1}, a_{t-1}) &= (1 - \eta_t) Q_{t-1}(s_{t-1}, a_{t-1}) + \eta_t \mathcal{T}_t(Q_{t-1})(s_{t-1}, a_{t-1}) \\ Q_t(s, a) &= Q_{t-1}(s, a), \qquad \forall (s, a) \neq (s_{t-1}, a_{t-1}) \end{aligned} \tag{6}$$

for any $t \geq 0$, whereas $\eta_t$ denotes the learning rate or the step size. Here $\mathcal{T}_t$ denotes the empirical Bellman operator w.r.t. the $t$-th sample, that is,

$$\mathcal{T}_t(Q)(s_{t-1}, a_{t-1}) := r(s_{t-1}, a_{t-1}) + \gamma \max_{a' \in \mathcal{A}} Q(s_t, a'). \tag{7}$$

It is worth emphasizing that at each time $t$, only a single entry — the one corresponding to the sampled state-action pair $(s_{t-1}, a_{t-1})$ — is updated, with all remaining entries unaltered. While the estimate $Q_0$ can be initialized to arbitrary values, we shall set $Q_0(s, a) = 0$ for all $(s, a)$ unless otherwise noted. The corresponding value function estimate $V_t : \mathcal{S} \to \mathbb{R}$ at time $t$ is thus given by

$$\forall s \in \mathcal{S}: \qquad V_t(s) := \max_{a \in \mathcal{A}} Q_t(s, a). \tag{8}$$

The complete algorithm is described in Algorithm 1.

---

**Algorithm 1:** Asynchronous Q-learning

---

**1 input parameters:** learning rates $\{\eta_t\}$, number of iterations $T$.
**2 initialization:** $Q_0 = 0$.
**3 for** $t = 1, 2, \cdots, T$ **do**
**4**     Draw action $a_{t-1} \sim \pi_{\mathsf{b}}(s_{t-1})$ and next state $s_t \sim P(\cdot | s_{t-1}, a_{t-1})$.
**5**     Update $Q_t$ according to (6).

---

**Theoretical guarantees for asynchronous Q-learning.** We are in a position to present our main theory regarding the non-asymptotic sample complexity of asynchronous Q-learning, for which the key parameters $\mu_{\mathsf{min}}$ and $t_{\mathsf{mix}}$ defined respectively in (3) and (4) play a vital role. The proof of this result is deferred to the full version [**?** ].

**Theorem 1** (Asynchronous Q-learning). *For the asynchronous Q-learning algorithm detailed in Algorithm 1, there exist some universal constants $c_0, c_1 > 0$ such that for any $0 < \delta < 1$ and $0 < \varepsilon \leq \frac{1}{1-\gamma}$, one has*

$$\forall (s, a) \in \mathcal{S} \times \mathcal{A}: \qquad |Q_T(s, a) - Q^\star(s, a)| \leq \varepsilon$$

*with probability at least $1 - \delta$, provided the iteration number $T$ and the learning rates $\eta_t \equiv \eta$ obey*

$$T \geq \frac{c_0}{\mu_{\mathsf{min}}} \left\{ \frac{1}{(1-\gamma)^5 \varepsilon^2} + \frac{t_{\mathsf{mix}}}{1-\gamma} \right\} \log\left(\frac{|\mathcal{S}||\mathcal{A}|T}{\delta}\right) \log\left(\frac{1}{(1-\gamma)^2 \varepsilon}\right), \qquad (9a)$$

$$\eta = \frac{c_1}{\log\left(\frac{|\mathcal{S}||\mathcal{A}|T}{\delta}\right)} \min\left\{ \frac{(1-\gamma)^4 \varepsilon^2}{\gamma^2}, \frac{1}{t_{\mathsf{mix}}} \right\}. \qquad (9b)$$

Theorem 1 delivers a finite-sample/finite-time analysis of asynchronous Q-learning, given that a fixed learning rate is adopted and chosen appropriately. The $\ell_\infty$-based sample complexity required for Algorithm 1 to attain $\varepsilon$ accuracy is at most

$$\widetilde{O}\left( \frac{1}{\mu_{\mathsf{min}}(1-\gamma)^5 \varepsilon^2} + \frac{t_{\mathsf{mix}}}{\mu_{\mathsf{min}}(1-\gamma)} \right). \qquad (10)$$

A few implications are in order.

**Dependency on the minimum state-action occupancy probability $\mu_{\mathsf{min}}$.** Our sample complexity bound (10) scales linearly in $1/\mu_{\mathsf{min}}$, which is in general unimprovable. Consider, for instance, the ideal scenario where state-action occupancy is nearly uniform across all state-action pairs, in which case $1/\mu_{\mathsf{min}}$ is on the order of $|\mathcal{S}||\mathcal{A}|$. In such a "near-uniform" case, the sample complexity scales linearly with $|\mathcal{S}||\mathcal{A}|$, and this dependency matches the known minimax lower bound [3] derived for the setting with independent samples. In comparison, [35, Theorem 7] depends at least quadratically on $1/\mu_{\mathsf{min}}$, which is at least $|\mathcal{S}||\mathcal{A}|$ times larger than our result (10).

**Dependency on the discount complexity $\frac{1}{1-\gamma}$.** The sample size bound (10) scales as $\frac{1}{(1-\gamma)^5 \varepsilon^2}$, which coincides with both [9, 44] (for the synchronous setting) and [5, 35] (for the asynchronous setting) with either a rescaled linear learning rate or a constant learning rate. This turns out to be the sharpest scaling known to date for the classical form of Q-learning.

**Dependency on the mixing time $t_{\mathsf{mix}}$.** The second additive term of our sample complexity (10) depends linearly on the mixing time $t_{\mathsf{mix}}$ and is (almost) independent of the target accuracy $\varepsilon$. The influence of this mixing term is a consequence of the expense taken for the Markovian trajectory to reach a steady state, which is a one-time cost that can be amortized over later iterations if the algorithm is run for reasonably long. Put another way, if the behavior chain mixes not too slowly with respect to $\varepsilon$ (in the sense that $t_{\mathsf{mix}} \leq \frac{1}{(1-\gamma)^4 \varepsilon^2}$), then the algorithm behaves as if the samples were independently drawn from the stationary distribution of the trajectory. In comparison, the influences of $t_{\mathsf{mix}}$ and $\frac{1}{(1-\gamma)^5 \varepsilon^2}$ in [35] (cf. Table 1) are multiplicative regardless of the value of $\varepsilon$, thus resulting in a much higher sample complexity.

**Schedule of learning rates.** An interesting aspect of our analysis lies in the adoption of a time-invariant learning rate, under which the $\ell_\infty$ error decays linearly — down to some error floor whose value is dictated by the learning rate. Therefore, a desired statistical accuracy can be achieved by

properly setting the learning rate based on the target accuracy level $\varepsilon$ and then determining the sample complexity accordingly. In comparison, classical analyses typically adopted a (rescaled) linear or a polynomial learning rule [20, 35]. While the work [5] studied Q-learning with a constant learning rate, their bounds were conservative and fell short of revealing the optimal scaling. Further, we note that adopting time-invariant learning rates is not the only option that enables the advertised sample complexity; as we shall elucidate shortly, one can also adopt carefully designed diminishing learning rates to achieve the same performance guarantees.

**Sample complexity based on the cover time.**    In addition, our analysis framework immediately leads to another sample complexity guarantee stated in terms of the cover time $t_{\text{cover}}$ (cf. (5)), which facilitates comparisons with several past work [5, 20].

**Theorem 2.** *For the asynchronous Q-learning algorithm detailed in Algorithm 1, there exist some universal constants $c_0, c_1 > 0$ such that for any $0 < \delta < 1$ and $0 < \varepsilon \leq \frac{1}{1-\gamma}$, one has*

$$\forall (s, a) \in \mathcal{S} \times \mathcal{A}: \qquad |Q_T(s, a) - Q^\star(s, a)| \leq \varepsilon$$

*with probability at least $1 - \delta$, provided the iteration number $T$ and the learning rates $\eta_t \equiv \eta$ obey*

$$T \geq \frac{c_0 t_{\text{cover}}}{(1-\gamma)^5 \varepsilon^2} \log^2 \left( \frac{|\mathcal{S}||\mathcal{A}|T}{\delta} \right) \log \left( \frac{1}{(1-\gamma)^2 \varepsilon} \right), \tag{11a}$$

$$\eta = \frac{c_1}{\log \left( \frac{|\mathcal{S}||\mathcal{A}|T}{\delta} \right)} \min \left\{ \frac{(1-\gamma)^4 \varepsilon^2}{\gamma^2}, 1 \right\}. \tag{11b}$$

In a nutshell, this theorem tells us that the $\ell_\infty$-based sample complexity of classical asynchronous Q-learning is bounded above by $\widetilde{O}\left( \frac{t_{\text{cover}}}{(1-\gamma)^5 \varepsilon^2} \right)$, which scales linearly with the cover time. This improves upon the prior result [20] (resp. [5]) by an order of at least $t_{\text{cover}}^{3.29} \geq |\mathcal{S}|^{3.29} |\mathcal{A}|^{3.29}$ (resp. $t_{\text{cover}}^2 |\mathcal{S}||\mathcal{A}| \geq |\mathcal{S}|^3 |\mathcal{A}|^3$). See Table 1 for detailed comparisons. We also make note of some connections between $t_{\text{cover}}$ and $t_{\text{mix}}/\mu_{\text{min}}$ to help compare Theorems 1-2: (1) in general, $t_{\text{cover}} = \widetilde{O}(t_{\text{mix}}/\mu_{\text{min}})$ for uniformly ergodic chains; (2) one can find some cases where $t_{\text{mix}}/\mu_{\text{min}} = \widetilde{O}(t_{\text{cover}})$. See [? , Appendix B] for more discussions.

**Adaptive and data-driven learning rates.**    The careful reader might remark that the learning rates recommended in (9b) depend on the mixing time $t_{\text{mix}}$ — a parameter that might be either *a priori* unknown or difficult to estimate. Fortunately, it is feasible to adopt a more adaptive learning rate schedule which does not rely on prior knowledge of $t_{\text{mix}}$ and which is still capable of achieving the performance advertised in Theorem 1.

In order to describe our new learning rate schedule, we need to keep track of the following quantities for all $(s, a) \in \mathcal{S} \times \mathcal{A}$:

- $K_t(s, a)$: the number of times that the sample trajectory visits $(s, a)$ during the first $t$ iterations.

In addition, we maintain an estimate $\widehat{\mu}_{\text{min}, t}$ of $\mu_{\text{min}}$, computed recursively as follows

$$\widehat{\mu}_{\text{min}, t} = \begin{cases} \frac{1}{|\mathcal{S}||\mathcal{A}|}, & \min_{s, a} K_t(s, a) = 0; \\ \widehat{\mu}_{\text{min}, t-1}, & \frac{1}{2} < \frac{\min_{s, a} K_t(s, a)/t}{\widehat{\mu}_{\text{min}, t-1}} < 2; \\ \min_{s, a} K_t(s, a)/t, & \text{otherwise}. \end{cases} \tag{12}$$

With the above quantities in place, we propose the following learning rate schedule:

$$\eta_t = \min \left\{ 1, c_\eta \exp \left( \left\lfloor \log \frac{\log t}{\widehat{\mu}_{\text{min}, t}(1-\gamma)\gamma^2 t} \right\rfloor \right) \right\}, \tag{13}$$

where $c_\eta > 0$ is some proper constant, and $\lfloor x \rfloor$ denotes the nearest integer less than or equal to $x$. If $\widehat{\mu}_{\text{min}, t}$ forms a reliable estimate of $\mu_{\text{min}}$, then one can view (13) as a sort of "piecewise constant approximation" of the rescaled linear stepsizes $\frac{c_\eta \log t}{\mu_{\text{min}}(1-\gamma)\gamma^2 t}$. Clearly, such learning rates are fully data-driven and do no rely on any prior knowledge about the Markov chain (like $t_{\text{mix}}$ and $\mu_{\text{min}}$) or the target accuracy $\varepsilon$.

---

**Algorithm 2:** Asynchronous variance-reduced Q-learning

---

1 **input parameters:** number of epochs $M$, epoch length $t_{\mathsf{epoch}}$, recentering length $N$, learning rate $\eta$.

2 **initialization:** set $Q_0^{\mathsf{epoch}} \leftarrow 0$.

3 **for** each epoch $m = 1, \cdots, M$ **do**

   /* Call Algorithm 3.                                                         */

4   $\quad Q_m^{\mathsf{epoch}} = \text{VR-Q-RUN-EPOCH}(\, Q_{m-1}^{\mathsf{epoch}}, N, t_{\mathsf{epoch}} )$ .

---

Encouragingly, our theoretical framework can be extended without difficulty to accommodate this adaptive learning rate choice. Specifically, for the Q-function estimates

$$\widehat{Q}_t = \begin{cases} Q_t, & \text{if } \eta_{t+1} \neq \eta_t, \\ \widehat{Q}_{t-1}, & \text{otherwise,} \end{cases} \tag{14}$$

we have the following theoretical guarantees.

**Theorem 3.** *Consider asynchronous Q-learning with learning rates* (13). *There is some sufficiently large universal constant $C > 0$ such that: for any $0 < \delta < 1$ and $0 < \varepsilon \leq \frac{1}{1-\gamma}$, one has*

$$\forall (s, a) \in \mathcal{S} \times \mathcal{A}: \qquad \big| \widehat{Q}_T(s, a) - Q^\star(s, a) \big| \leq \varepsilon \tag{15}$$

*with probability at least $1 - \delta$, provided that*

$$T \geq C \max \left\{ \frac{1}{\mu_{\mathsf{min}}(1-\gamma)^5 \varepsilon^2}, \frac{t_{\mathsf{mix}}}{\mu_{\mathsf{min}}(1-\gamma)} \right\} \log \left( \frac{|\mathcal{S}||\mathcal{A}|T}{\delta} \right) \log \left( \frac{T}{(1-\gamma)^2 \varepsilon} \right). \tag{16}$$

## 4   Extension: asynchronous variance-reduced Q-learning

As pointed out in prior literature, the classical form of Q-learning (6) often suffers from sub-optimal dependence on the discount complexity $\frac{1}{1-\gamma}$. For instance, in the synchronous setting, the minimax lower bound is proportional to $\frac{1}{(1-\gamma)^3}$ (see, [3]), while the sharpest known upper bound for vanilla Q-learning scales as $\frac{1}{(1-\gamma)^5}$; see detailed discussions in [44]. To remedy this issue, recent work proposed to leverage the idea of variance reduction to develop accelerated RL algorithms in the synchronous setting [37, 45], as inspired by the seminal SVRG algorithm [26]. These prior results, however, focused on the synchronous setting with independent samples. In this section, we adapt this idea to asynchronous Q-learning and characterize its sample efficiency.

**Algorithm.**   In order to accelerate the convergence, it is instrumental to reduce the variability of the empirical Bellman operator $\mathcal{T}_t$ employed in the update rule (6) of classical Q-learning. This can be achieved via the following means. Simply put, assuming we have access to *(i)* a reference Q-function estimate, denoted by $\overline{Q}$, and *(ii)* an estimate of $\mathcal{T}(\overline{Q})$, denoted by $\widetilde{\mathcal{T}}(\overline{Q})$, the variance-reduced Q-learning update rule is given by

$$Q_t(s_{t-1}, a_{t-1}) = (1 - \eta_t) Q_{t-1}(s_{t-1}, a_{t-1}) + \eta_t \Big( \mathcal{T}_t(Q_{t-1}) - \mathcal{T}_t(\overline{Q}) + \widetilde{\mathcal{T}}(\overline{Q}) \Big)(s_{t-1}, a_{t-1}),$$
$$Q_t(s, a) = Q_{t-1}(s, a), \qquad \forall (s, a) \neq (s_{t-1}, a_{t-1}), \tag{17}$$

where $\mathcal{T}_t$ denotes the empirical Bellman operator at time $t$ (cf. (7)). The empirical estimate $\widetilde{\mathcal{T}}(\overline{Q})$ can be computed using a set of samples; more specifically, by drawing $N$ consecutive sample transitions $\{(s_i, a_i, s_{i+1})\}_{0 \leq i < N}$ from the observed trajectory, we compute

$$\widetilde{\mathcal{T}}(\overline{Q})(s, a) = r(s, a) + \frac{\gamma \sum_{i=0}^{N-1} \mathbb{1}\{(s_i, a_i) = (s, a)\} \max_{a'} \overline{Q}(s_{i+1}, a')}{\sum_{i=0}^{N-1} \mathbb{1}\{(s_i, a_i) = (s, a)\}}. \tag{18}$$

Compared with the classical form (6), the original update term $\mathcal{T}_t(Q_{t-1})$ has been replaced by $\mathcal{T}_t(Q_{t-1}) - \mathcal{T}_t(\overline{Q}) + \widetilde{\mathcal{T}}(\overline{Q})$, in the hope of achieving reduced variance as long as $\overline{Q}$ (which serves as a proxy to $Q^\star$) is chosen properly.

---

**Algorithm 3:** function $Q = \text{VR-Q-RUN-EPOCH}(\overline{Q}, N, t_{\text{epoch}})$

---

1   Draw $N$ new consecutive samples from the sample trajectory; compute $\widetilde{\mathcal{T}}(\overline{Q})$ with (18).
2   Set $s_0 \leftarrow$ current state, and $Q_0 \leftarrow \overline{Q}$.
3   **for** $t = 1, 2, \cdots, t_{\text{epoch}}$ **do**
4      Draw action $a_{t-1} \sim \pi_{\text{b}}(s_{t-1})$ and next state $s_t \sim P(\cdot|s_{t-1}, a_{t-1})$.
5      Update $Q_t$ according to (17).
6   **return:** $Q \leftarrow Q_{t_{\text{epoch}}}$.

---

For convenience of presentation, we introduce the following notation

$$Q = \text{VR-Q-RUN-EPOCH}(\overline{Q}, N, t_{\text{epoch}}) \tag{19}$$

to represent the above-mentioned update rule, which starts with a reference point $\overline{Q}$ and operates upon a total number of $N + t_{\text{epoch}}$ consecutive sample transitions. The first $N$ samples are employed to construct $\widetilde{\mathcal{T}}(\overline{Q})$ via (18), with the remaining samples employed in $t_{\text{epoch}}$ iterative updates (17); see Algorithm 3. To achieve the desired acceleration, the proxy $\overline{Q}$ needs to be periodically updated so as to better approximate the truth $Q^\star$ and hence reduce the bias. It is thus natural to run the algorithm in a multi-epoch manner. Specifically, we divide the samples into contiguous subsets called epochs, each containing $t_{\text{epoch}}$ iterations and using $N + t_{\text{epoch}}$ samples. We then proceed as follows

$$Q_m^{\text{epoch}} = \text{VR-Q-RUN-EPOCH}(Q_{m-1}^{\text{epoch}}, N, t_{\text{epoch}}), \quad m = 1, \ldots, M, \tag{20}$$

where $M$ is the total number of epochs, and $Q_m^{\text{epoch}}$ denotes the output of the $m$-th epoch. The whole procedure is summarized in Algorithm 2. Clearly, the total number of samples used in this algorithm is given by $M(N + t_{\text{epoch}})$. We remark that the idea of performing variance reduction in RL is certainly not new, and has been explored in a number of recent work [16, 29, 37, 38, 45, 50].

**Theoretical guarantees for variance-reduced Q-learning.** We develop a non-asymptotic sample complexity bound for asynchronous variance-reduced Q-learning on a single trajectory. Before presenting our theoretical guarantees, there are several algorithmic parameters that we shall specify; for given target levels $(\varepsilon, \delta)$, choose

$$\eta_t \equiv \eta = \frac{c_0}{\log\left(\frac{|\mathcal{S}||\mathcal{A}|t_{\text{epoch}}}{\delta}\right)} \min\left\{\frac{(1-\gamma)^2}{\gamma^2}, \frac{1}{t_{\text{mix}}}\right\}, \tag{21a}$$

$$N \geq \frac{c_1}{\mu_{\text{min}}}\left(\frac{1}{(1-\gamma)^3 \min\{1, \varepsilon^2\}} + t_{\text{mix}}\right) \log\left(\frac{|\mathcal{S}||\mathcal{A}|t_{\text{epoch}}}{\delta}\right), \tag{21b}$$

$$t_{\text{epoch}} \geq \frac{c_2}{\mu_{\text{min}}}\left(\frac{1}{(1-\gamma)^3} + \frac{t_{\text{mix}}}{1-\gamma}\right) \log\left(\frac{1}{(1-\gamma)^2 \varepsilon}\right) \log\left(\frac{|\mathcal{S}||\mathcal{A}|t_{\text{epoch}}}{\delta}\right), \tag{21c}$$

where $c_0 > 0$ is some sufficiently small constant, $c_1, c_2 > 0$ are some sufficiently large constants, and we recall the definitions of $\mu_{\text{min}}$ and $t_{\text{mix}}$ in (3) and (4), respectively. Note that the learning rate (21a) chosen here could be larger than the choice (9b) for the classical form by a factor of $O\left(\frac{1}{(1-\gamma)^2}\right)$ (which happens if $t_{\text{mix}}$ is not too large), allowing the algorithm to progress more aggressively.

**Theorem 4** (Asynchronous variance-reduced Q-learning). *Let $Q_M^{\text{epoch}}$ be the output of Algorithm 2 with parameters chosen according to (21). There exists some constant $c_3 > 0$ such that for any $0 < \delta < 1$ and $0 < \varepsilon \leq \frac{1}{1-\gamma}$, one has*

$$\forall(s, a) \in \mathcal{S} \times \mathcal{A}: \qquad |Q_M^{\text{epoch}}(s, a) - Q^\star(s, a)| \leq \varepsilon$$

*with probability at least $1 - \delta$, provided that the total number of epochs exceeds*

$$M \geq c_3 \log \frac{1}{\varepsilon(1-\gamma)^2}. \tag{22}$$

In view of Theorem 4, the $\ell_\infty$-based sample complexity for variance-reduced Q-learning to yield $\varepsilon$ accuracy — which is characterized by $M(N + t_{\text{epoch}})$ — can be as low as

$$\widetilde{O}\left(\frac{1}{\mu_{\text{min}}(1-\gamma)^3 \min\{1, \varepsilon^2\}} + \frac{t_{\text{mix}}}{\mu_{\text{min}}(1-\gamma)}\right). \tag{23}$$

Except for the second term that depends on the mixing time, the first term matches [45] derived for the synchronous settings with independent samples. In the range $\varepsilon \in (0, \min\{1, \frac{1}{(1-\gamma)\sqrt{t_{\mathsf{mix}}}}\}]$, the sample complexity reduce to $\widetilde{O}\big(\frac{1}{\mu_{\mathsf{min}}(1-\gamma)^3\varepsilon^2}\big)$; the scaling $\frac{1}{(1-\gamma)^3}$ matches the minimax lower bound derived in [3] for the synchronous setting.

## 5  Related work

**The Q-learning algorithm and its variants.**  The Q-learning algorithm, originally proposed in [47], has been analyzed in the asymptotic regime by [7, 24, 42, 43] since more than two decades ago. Additionally, finite-time performance of Q-learning and its variants have been analyzed by [5, 9, 20, 28, 35, 44, 48] in the tabular setting, by [6, 8, 10, 17, 18, 21, 49, 52] in the context of function approximations, and by [36] with nonparametric regression. In addition, [2, 14, 22, 37, 40, 45] studied modified Q-learning algorithms that might potentially improve sample complexities and accelerate convergence.

**Finite-sample $\ell_\infty$ guarantees for Q-learning.**  We now expand on non-asymptotic $\ell_\infty$ guarantees available in prior literature, which are the most relevant to the current work. An interesting aspect that we shall highlight is the importance of learning rates. For instance, when a linear learning rate (i.e. $\eta_t = 1/t$) is adopted, the sample complexity results derived in past work [20, 42] exhibit an exponential blow-up in $\frac{1}{1-\gamma}$, which is clearly undesirable. In the synchronous setting, [5, 9, 20, 44] studied the finite-sample complexity of Q-learning under various learning rate rules; the best sample complexity known to date is $\widetilde{O}\big(\frac{|\mathcal{S}||\mathcal{A}|}{(1-\gamma)^5\varepsilon^2}\big)$, achieved via either a rescaled linear learning rate [9, 44] or a constant learning rate [9]. When it comes to asynchronous Q-learning (in its classical form), our work provides the first analysis that achieves linear scaling with $1/\mu_{\mathsf{min}}$ or $t_{\mathsf{cover}}$; see Table 1 for detailed comparisons. Going beyond classical Q-learning, the speedy Q-learning algorithm provably achieves a sample complexity of $\widetilde{O}\big(\frac{t_{\mathsf{cover}}}{(1-\gamma)^4\varepsilon^2}\big)$ [2] in the asynchronous setting, whose update rule takes twice the storage of classical Q-learning. In comparison, our analysis of the variance-reduced Q-learning algorithm achieves a sample complexity of $\widetilde{O}\big(\frac{1}{\mu_{\mathsf{min}}(1-\gamma)^3\varepsilon^2} + \frac{t_{\mathsf{mix}}}{\mu_{\mathsf{min}}(1-\gamma)}\big)$ when $\varepsilon < 1$.

**Finite-sample guarantees for model-free algorithms.**  Convergence of several model-free RL algorithms has been studied recently in the presence of Markovian data, including but not limited to TD learning and its variants [6, 11, 12, 15, 23, 27, 30, 39, 50, 51], Q-learning [10, 49], and SARSA [53]. However, these recent papers typically focused on the (weighted) $\ell_2$ error rather than the $\ell_\infty$ risk, where the latter is often more relevant in the context of RL. In addition, [29, 33] investigated the $\ell_\infty$ bounds of (variance-reduced) TD learning, although they did not account for Markovian noise.

**Finite-sample guarantees for model-based algorithms.**  Another contrasting approach for learning the optimal Q-function is the class of model-based algorithms, which has been shown to enjoy minimax-optimal sample complexity $\widetilde{O}\big(\frac{|\mathcal{S}||\mathcal{A}|}{(1-\gamma)^3\varepsilon^2}\big)$ in the synchronous setting [1, 3, 31]. It is worth emphasizing that the minimax optimality of model-based approach has been shown to hold for the entire $\varepsilon$-range; in comparison, the sample optimality of the model-free approach has only been shown for a smaller range of accuracy level $\varepsilon$ in the synchronous setting.

## 6  Discussion

This work develops a sharper finite-sample analysis of the classical asynchronous Q-learning algorithm, highlighting and refining its dependency on intrinsic features of the Markovian trajectory induced by the behavior policy. Our sample complexity bound strengthens the state-of-the-art result by an order of at least $|\mathcal{S}||\mathcal{A}|$. A variance-reduced variant of asynchronous Q-learning is also analyzed, exhibiting improved scaling with the discount complexity $\frac{1}{1-\gamma}$.

Our findings and the analysis framework developed herein suggest a couple of directions for future investigation. For instance, our improved sample complexity of asynchronous Q-learning has a dependence of $\frac{1}{(1-\gamma)^5}$ on the discount complexity, which is inferior to its model-based counterpart. In the synchronous setting, [44] demonstrated an empirical lower bound $\frac{1}{(1-\gamma)^4}$ for Q-learning. It would be important to determine the exact scaling in this regard. In addition, it would be interesting to see whether the techniques developed herein can be exploited towards understanding model-free algorithms with more sophisticated exploration schemes [4, 13, 25].

## Broader Impact

This work is a theoretical contribution to characterize the sample complexity of asynchronous Q-learning. The insights from the proposed algorithm can potentially be leveraged in various reinforcement learning tasks in the future.

## Acknowledgments and Disclosure of Funding

G. Li and Y. Gu are supported in part by the grant NSFC-61971266. Y. Wei is supported in part by the grants NSF CCF-2007911 and DMS-2015447. Y. Chi is supported in part by the grants ONR N00014-18-1-2142 and N00014-19-1-2404, ARO W911NF-18-1-0303, and NSF CCF-1806154 and CCF-2007911. Y. Chen is supported in part by the grants AFOSR YIP award FA9550-19-1-0030, ONR N00014-19-1-2120, ARO YIP award W911NF-20-1-0097, ARO W911NF-18-1-0303, NSF CCF-1907661, DMS-2014279 and IIS-1900140.

## Footnotes

[1] Let $\mathcal{X} := \left(\lvert\mathcal{S}\rvert, \lvert\mathcal{A}\rvert, \frac{1}{1-\gamma}, \frac{1}{\varepsilon}\right)$. The notation $f(\mathcal{X}) = O(g(\mathcal{X}))$ means there exists a universal constant $C_1 > 0$ such that $f \le C_1 g$. The notation $\widetilde{O}(\cdot)$ is defined analogously except that it hides any logarithmic factor.

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
