[Reviews · NeurIPS 2020]

Review 1

Summary and Contributions: The paper provides a finite-time analysis of the convergence of asynchronous Q-learning in the infinite-horizon discounted RL setting, limited to tabular MDPs. The authors prove that a particular choice of a constant learning rate allows achieving a sample complexity of order \tilde{O}((1-\gamma)^{-5}\epsilon^2). Moreover, the authors show that leveraging to variance-reduction techniques it is possible to further improve to \tilde{O}((1-\gamma)^{-3}\epsilon^2).

Strengths: The paper is exclusively theoretical. Although it can be considered an incremental work compared for instance to the recent [33], I think that the paper has some points of strength: 1. The dependence on (1-\gamma) and \epsilon are the same as in [33], but the dependence on \mu_{min} (the minimum entry of the stationary distribution induced by the exploration policy) and t_{mix} (the mixing time) is improved compared to [33]. 2. The extension to variance-reduction techniques allows further improving the sample complexity. The application of these techniques to the asynchronous Q-learning is novel. [33] Guannan Qu and Adam Wierman. Finite-time analysis of asynchronous stochastic approximation and Q-learning. accepted to Conference on Learning Theory, 2020.

Weaknesses: My main concern about the paper is whether this proposed algorithm is actually implementable due to the specific expression of the (constant) learning rate. I have two concerns: 1. The learning rate depends on t_{mix} in Theorem 1 and on the universal constants c_1 in both Theorem 1 and Theorem 2. How can we compute/approximate t_{mix} in advance? If we cannot, is it sufficient to employ a lower-bound on t_{mix}? Concerning the constant c_1. Looking at the proofs c_1 is a function of constant c (Equation 55) that in turn derives from Bernstein's inequality (Equation 81) and subsequently \tilde{c} (Equation 84), but its value is never explicitly computed. I am aware that also in [33] the learning rate schedule (that is not constant) depends on \mu_{min} and t_{mix}, but I think the authors should elaborate more on this and explain how to deal with it in practice, if possible. Furthermore, in [19] this dependence on t_{mix} is not present as the learning rate is chosen as a power of 1/n(s,a), where n(s,a) is the number of visits to state-action pair (s,a). In [19] the theoretical guarantees are weaker. I wonder if using a learning rate independent from t_{mix} prevents from achieving these stronger convergence guarantees. 2. Provided that we can deal with the issues of the previous point, a deeper problem is that the learning rate depends on the accuracy threshold \epsilon and on the number of iterations T. This suggests that a hypothetical algorithm should decide in advance \epsilon and T (maybe a user input) and then compute the learning rate. If we let the algorithm proceed to learn beyond T what will happen? Will it converge asymptotically? Intuitively, looking at the expression of the learning rate, to have asymptotic convergence I should set T->\infty and \epsilon->0 leading to a learning rate of value zero. It seems to me that this issue is specific to this work and not present for instance in [33]. If I look at Theorem 7 of [33], the learning rate schedule does not depend on T and the bound can be instanced for any T. Thus, for T->infty we will have asymptotic convergence. If I am correct on this reasoning, I think that the algorithm has a non-negligible limitation that prevents it from being applied in practice. Can the authors clarify this point? [19] Eyal Even-Dar and Yishay Mansour. Learning rates for Q-learning. Journal of machine learning Research, 5(Dec):1–25, 2003. ***Minor*** - There are some broken references in Appendix B - There are some equations in Appendix that go beyond the margin - In Appendix C, the definition of t_{cover} (Equation 34a) is different from the definition of t_{cover} presented in the main paper (Equation 5). I suggest employing different symbols to avoid confusion.

Correctness: I made an overall check of the proofs in the appendix and the proofs seem correct to me and relatively easy to follow.

Clarity: The paper is written in good English and reads well.

Relation to Prior Work: There is an adequate related works section in which the proposed approach is compared with state-of-the-art methods.

Reproducibility: Yes

Additional Feedback: I think that the paper has good potential. However, I have some concerns about the expression of the learning rate and on the possibility that it can be computed to obtain an implementable algorithm. I am willing to increase my score provided that the authors properly address my concerns. ***Post-Rebuttal Feedback*** I have read the author's feedback together with the other reviews. I appreciate the proposal of an implementable version of the learning rate and the corresponding theoretical analysis. I think the rebuttal solves my concerns about the learning rate expression. In view of this, I am happy to raise my score.


Review 2

Summary and Contributions: In this paper, the authors derive non-asymptotic bounds for asynchronous Q-learning. Further, they also analyze a variant of Q-learning that incorporates variance reduction.

Strengths: Improved sample complexity bounds

Weaknesses: No discussion of a problematic step-size dependence

Correctness: I did not check the proofs

Clarity: The paper is well-written

Relation to Prior Work: Yes, the comparison to bounds in previous works is clear.

Reproducibility: Yes

Additional Feedback: In this paper, the authors derive non-asymptotic bounds for asynchronous Q-learning. Further, they also analyze a variant of Q-learning that incorporates variance reduction. The paper is well-written, and Im going with a positive score for this submission. However, I have one major grievance concerning a step-size dependence. The bound in Theorem 1 is for a step-size that requires knowledge of the mixing time of the behaviour policy, and this information is not available in a typical RL setting. At the very least, I would like to see a discussion on this step-size dependence. What is the motivation for using a constant step-size? Or, why not use a diminishing step-size? The motivation for this query is a constraint that requires the step-size to be sufficiently small (eta< 1/t_mix). Such a constraint could be satisfied for a diminishing step-size, say eta_t, for sufficiently large t. While the bound in Theorem 1 exhibits better dependence on t_mix through an additive form, the bound in a previous work, such as [19], is for a step-size that doesnt require knowledge of t_mix. From Theorem C.1 in the appendix, is there an alternative possibility for the choice of \eta, that gets rid of t_mix dependence? Or, does iterate averaging help? On the bound using t_cover in Table 1: I see that the step-size setting here is practical as mixing time of the underlying Markov chain isnt required. As per my understanding, the bound in the last row isn't necessarily better than that in the first row, as t_cover and the discount factor would both play a role in deciding. While I did not check the proofs, Im curious to know if bounds in expectation can be extracted from the proofs for sample complexity bounds. --------------- Update after author feedback: I have checked the author feedback, and will make an upward revision of my score to vote for acceptance. Some notes: 1. The main grievance was concerning the step-size choice, which required knowledge of underlying transition dynamics. The author feedback addressed this issues. 2. The authors haven't clarified on a couple of my queries: (i) why not use a diminishing step-size; and (ii) The bound using t_cover doesn't appear to be always better than that in [19].


Review 3

Summary and Contributions: This paper develops a refined analysis framework about the sample efficiency of classical asynchronous Q-learning on a single sample trajectory. This article focuses on the asynchronous learning method for Q-Learning, and gives the theoretical bound of the epsilon optimal Q-function estimate. In addition, the author leverage the idea of variance reduction to improve the scaling with the discount complexity and proposed a variation-reduced variant of asynchronous Q-learning method. Given the constant learning rate, the bound of Sample complexity of this article can be derived.

Strengths: The advantage of this article is that it provides the theoretical bound of Sample complexity of asynchronous Q-learning. I think this is very meaningful, because most of the Q-Learning and RL algorithms lack theoretical guarantees, and they are more sample inefficient, requiring a lot of sampling to make the learned Q function converge. Therefore, the author obtained the bound of sample complexity of asynchronous Q-learning through theoretical analysis, which I think is very meaningful for the development of Q-Learning.

Weaknesses: I think the shortcoming of this article is that it does not clearly explain the methods and defects of asynchronous Q-learning. I read many other articles including the asynchronous one-step Q-learning and asynchronous n-steps Q-learning methods in the A3C article to understand the background problem and direction that the author generally wants to solve. Also, the related articles given in this article basically did not mention the asynchronous Q-learning method, so I think the description part of this article needs some improvement.

Correctness: I tried to understand and follow the theoretical analysis part of the article, including the supplementary material part, but I have to admit that I did not follow some theoretical parts. I think this article is more theoretical, so if my review comments are a bit biased, I hope other reviewers could understand.

Clarity: Yes, this article is well organized.

Relation to Prior Work: Yes, the author clearly stated the contribution of this article and the shortcomings of the previous methods

Reproducibility: Yes

Additional Feedback:


Review 4

Summary and Contributions: This paper studies the sample complexity of the classical Q-learning algorithm in the asynchronous setting. The authors present new PAC-type sample complexities for asynchronous Q-learning (henceforth, Async-QL) sharpening existing bounds by a factor of t_mix/\mu_min asymptotically (namely, when the error \epsilon tends to zero). Here, t_mix and \mu_min respectively denote the mixing time and the minimum state-action occupancy probability of the Markov chain induced by the behavior policy used in Async-QL. This also leads to another sample complexity for Async-QL in terms of the cover time (t_cover) associated to the chain induced by the behavior policy. The two results improve upon recent best known results for Async-QL. The second contribution of the paper is to present a sample complexity bound for a variance-reduced variant of Async-QL, whose design is inspired by the variance-reduction techniques originally used to accelerate SGD. The paper shows that using technique leads to removal of a factor of 1/(1-\gamma)^2 from the main term in the sample complexity bounds for Async-QL discussed above.

Strengths: The first strong aspect of the paper is to study the performance of the “asynchronous” variant of QL, thereby relaxing the prevailing assumption of access to a generative model (a.k.a. the synchronous setting) in most related prior work. While Async-QL has been around for more than three decades, most literature, until very recently, focused on investigating the performance of synchronous QL (and its variants in the synchronous setting). The paper could be seen amongst very few works studying the performance of Async-QL. On the technical side, the first result (Theorem 1) improves on the best existing known sample complexity of Async-QL. The main term of this bound, which becomes dominant when \epsilon is small enough, sharpens the best existing bound by a factor of t_mix/\mu_min, which is a significant gain. Also it shows that the price to pay, compared to the synchronous setting, is an additive \epsilon-independent term, which is intuitive. Another strength of the paper is the bound in Theorem 2: while it could be looser than that in Theorem 1 by, at most, a factor of t_\mix, it improves on a former similar result in [4] by a factor of SA*t_cover, and more importantly, it is achieved by a learning rate independent of unknown environment-dependent quantities. Another technical strength of the paper is to remove further a factor of 1/(1-\gamma)^2 from the sample complexity through variance-reduction techniques. Last, a general strong aspect of the paper is that it is very well-written and well-organized, and admits a clear and precise presentation.

Weaknesses: The are some weaknesses associated to the bounds in Theorems 1 and 3. In particular, for Theorem 1 the authors silently assume that the learner knows t_mix or an upper bound on it: the sample complexity is achieved by a learning rate depending on t_mix (which is initially unknown). Unless I am missing something, this implies that a learner agnostic to t_mix cannot achieve the sample complexity of Theorem 1 unless \epsilon is small enough (relative to t_mix). Of course, I remark that asymptotically (when \epsilon goes to zero), one can partially circumvent this issue as the learning rate is effectively determined by \epsilon and \gamma. However, the learning rate still depends on 1/log(T), where T itself depends on t_mix/\mu_min. Let me emphasize that one can maintain high-probability confidence intervals for t_mix (and \mu_min) so as to safely pick a proper learning rate (in view of the result of Wolfer and Kontorovich (2019), I think this may not hurt much the sample complexity of Theorem 1). However, the resulting method will significantly differ from classical Async QL, and more importantly, it will no longer be model-free. The same argument applies to Theorem 3 as well, where the learner needs to know both t_mix and \mu_min -- or at least relevant bounds on them. A similar argument may also apply to the bound in Theorem 2: the learning rate depends on \epsilon and \gamma, and \log(T), where T itself depends on t_cover. Of course, this is less crucial compared to the situation of Theorems 1 and 3, as the dependence on unknown quantities appears only in logarithmic terms. I understand that a similar drawback may be true in some recent works too, as reported in Table 1. However, I believe this is a crucial drawback if one aims to truly address the fundamental question in line 43 -- where achievability is a considered. Unfortunately I was unable to find any related comment on this. I therefore would like to hear the authors’ clarifications on this, and in particular, whether (and to which extent) the dependence of learning rate on unknown quantities could be potentially relaxed. At the very least, it is expected that they clearly and precisely discuss this in the paper.

Correctness: The results presented in the paper look correct to me. I was unable to check all details in the appendix. However, after a quick reading of proofs, I highlight that they appear correct to me. Besides, there are some statements about superiority of model-free methods to model-based ones that sound rather incorrect (or at least, I disagree with): For example, in line 24, it is implied that model-free methods adapt flexibly to changing environments (and perhaps autonomously). First it is not clear how, e.g., QL is guaranteed to adapt efficiently to non-stationarity, and therefore, I urge to authors to provide a suitable reference. Further, adaptation to non-stationary environments is also possible in model-based RL algorithms. In view of the state-of-the-art methods based on change-point detection, I believe that the fastest and most efficient ways in doing so is to use model-based methods.

Clarity: The paper is overall very well-written and easy-to-follows. It admits a clear and precise presentation.

Relation to Prior Work: The paper cites almost all relevant literature that I am aware of. Furthermore, it precisely explains its contribution compare to existing works and results.

Reproducibility: Yes

Additional Feedback: The paper is very well-polished. However, I found a few typos reported below: l. 25: …. and …. of ... lies ==> … lie l. 76: classical literature ==> the classical literature l. 92 (and several other places): (in LaTeX) S \times A \mapsto [0,1] ==> …. \to ... ---- \mapsto is typically used when defining the actual rule of function or mapping (like x\mapsto \log(x)). I. 66: the statement is unclear to me. Does it mean “when \epsilon goes to zero?” Appendix B: t_mix and \tau_min seemingly refer to the same quantity, the mixing time. ==== AFTER REBUTTAL ==== I have read the author feedback and the other reviews. The authors provided convincing answers to my concerns, and I appreciate their effort given the short rebuttal period. As a result, and assuming that the authors will incorporate all these changes properly into the final version, I increase my score to 7.

[Author Response · NeurIPS 2020]

We thank the reviewers for very helpful comments. This letter addresses the **major questions** raised by the reviewers.

**Learning rates.** To address the reviewers' comments on learning rates, we will add results with *easy-to-implement*
*learning rates*, without compromising sample complexities. Specifically, for some constant $c > 0$ let

$$\eta_t = \min\left\{1, c\exp\left(\left\lfloor \log\frac{\log t}{\widehat{\mu}_{\min,t}(1-\gamma)\gamma^2 t}\right\rfloor\right)\right\} \qquad \text{(an epoch-based choice)} \qquad (1)$$

which can be viewed as a "piecewise approximation" of the rescaled linear stepsizes $\eta_t = \min\left\{1, \frac{c\log t}{\widehat{\mu}_{\min,t}(1-\gamma)\gamma^2 t}\right\}$.
Here, $\widehat{\mu}_{\min,t}$ is the minimum entry of certain empirical state-action visitation probability vector.[1]. Clearly, this choice
does *not* rely on the mixing time $t_{\mathsf{mix}}$, minimum state-action occupancy probability $\mu_{\min}$, and target accuracy $\varepsilon$.
Encouragingly, our current theory can be easily extended to cover this easier-to-implement learning rate choice:

**Theorem 5** ($\ell_\infty$ *sample complexity for achieving* $\varepsilon$ *accuracy*)**.** *Consider asynchronous Q-learning with learning*
*rates* (1). *There exists some universal constant* $C > 0$ *such that: for any* $0 < \delta < 1$ *and* $0 < \varepsilon \leq \frac{1}{1-\gamma}$, *one has*
$\|Q_T - Q^\star\|_\infty \leq \varepsilon$ *with probability at least* $1-\delta$, *provided that the sample size (or number of iterations)* $T$ *obeys*

$$T \geq C\max\left\{\frac{1}{\mu_{\min}(1-\gamma)^5\varepsilon^2}, \frac{t_{\mathsf{mix}}}{\mu_{\min}(1-\gamma)}\right\}\cdot\mathsf{polylog}\left(|\mathcal{S}|, |\mathcal{A}|, \frac{1}{1-\gamma}, T, \frac{1}{\delta}, \frac{1}{\varepsilon}\right). \qquad (2)$$

Similarly, our theory for variance-reduced Q-learning can also be extended to a stepsize that does not depend on $t_{\mathsf{mix}}$.
More specifically, this requires two changes: (1) the epoch length needs to keep increasing (i.e. at the end of every
epoch, run $t_{\mathsf{epoch}} \leftarrow 2t_{\mathsf{epoch}}$); (2) set $\eta_t = \frac{c\log t_{\mathsf{epoch}}}{\widehat{\mu}_{\min,t}(1-\gamma)t_{\mathsf{epoch}}}$. This can be analyzed via a similar argument.

*Proof of Theorem 5.* We sketch the proof for the piecewise choice (1), which follows easily from our Theorem 1.

1) Set $T_0 = T/2$. Given that $\eta_t \leq 1$, it is easily seen that $\|Q_{T_0} - Q^\star\|_\infty \lesssim T_0$. To simplify presentation, we assume
here that $T$ is the point where $\eta_t$ undergoes a change (we can easily cover general cases via epoch-based analysis).

2) Choose $\widetilde{\varepsilon}$ s.t. $\frac{\mu_{\min}(1-\gamma)T/2}{\log\left(\frac{|\mathcal{S}||\mathcal{A}|T/2}{\delta}\right)\log\frac{T}{2}} = \frac{C}{(1-\gamma)^4\widetilde{\varepsilon}^2}$, which obeys $\widetilde{\varepsilon} \leq \varepsilon$ under Condition (2). Combining the piecewise
choice (1) and Condition (2) implies: $\eta_t \equiv \frac{c'}{\log\left(\frac{|\mathcal{S}||\mathcal{A}|T}{\delta}\right)}\min\left\{(1-\gamma)^4\widetilde{\varepsilon}^2, \frac{1}{t_{\mathsf{mix}}}\right\}, \forall t \in [T_0, T]$, where $c'$ is some constant.

3) With the above learning rate condition in mind, invoking Theorem 1 of our paper with initialization $Q_{T_0}$ ensures that
$\|Q_T - Q^\star\|_\infty \leq \widetilde{\varepsilon} \leq \varepsilon$ with probability at least $1-\delta$, provided that the sample size condition (2) holds. $\square$

**Specific questions by Reviewer 1**: 1. *"Implementable learning rates"*: See our response above on "learning rates".
While the constant $c$ in $\eta_t$ can also be specified explicitly (by using specific constants in Bernstein inequality, etc),
we caution that such a theoretical constant might be overly conservative in practice, given that our theory focuses on
orderwise sample complexity bounds and does not strive to sharpen the constant. We will clarify this in the revision.

2. *"Main contributions"*: In comparison to the state-of-the-art [33] which unveiled tight scaling w.r.t. the important
factors $\frac{1}{1-\gamma}$ and $\frac{1}{\varepsilon}$, our main focus is towards sharpening the dependency on the problem dimension $|\mathcal{S}||\mathcal{A}|$ through
improving the dependency on $\mu_{\min}$. Specifically, we improve prior sample complexity bound by a factor of $1/\mu_{\min} \geq$
$|\mathcal{S}||\mathcal{A}|$. Given that $|\mathcal{S}||\mathcal{A}|$ is often enormous in practice, our theory potentially leads to a notable improvement.

**Specific questions by Reviewer 2**: 1. *"Dependency of stepsizes on $t_{\mathsf{mix}}$"*: See the response above on "learning rates".

2. *"Bounds in expectation"*. A bound on expectation can also be extracted by (1) using the boundedness nature of the
Q-update and (2) choosing $\delta$ to be sufficiently small. We will add this in the revision.

**Specific questions by Reviewer 3**: *"Asynchronous Q-learning vs. A3C"*: We'd like to clarify a possible source of
confusion due to the different use of terminology in two different topics. The word "asynchronous" in Q-learning
was often used in classical Q-learning literature (e.g. Tsitsiklis [41]) to indicate that: at every time only a single state-
action pair in the Q-function is updated. This is in stark contrast to another line of recent literature on asynchronous
optimization, which studies asynchronous updates of multiple CPU threads in parallel/multi-agent optimization (for
instance, the A3C paper uses asynchronous SGD to simultaneously deploy/coordinate multiple CPU threads). Hence,
the two settings are indeed quite different, although the Q-learning algorithm studied here might be used in any single
thread to perform some component of an RL algorithm. We will clarify this in the revision to avoid confusion.

**Specific questions by Reviewer 5**: 1. *"Dependency of stepsizes on $t_{\mathsf{mix}}$"*: See the response above on "learning rates".

2. *"Advantages of model-free methods:"* Thanks for raising concerns about our statements on model-based vs. model-
free RL. We will rephrase these statements in the revision based on the reviewer's suggestion.

## Footnotes

[1]Using concentration bounds in the supplement, we can ensure $\widehat{\mu}_{\min,t} = (1+o(1))\mu_{\min}$ for all $t \gtrsim t_{\mathsf{mix}}/\mu_{\min}$ (up to log factor).


[Meta-Review · NeurIPS 2020]

The reviewers appreciated the efforts made by the authors in the rebuttal, and updated their reviews accordingly. The paper contributions are now clear and important (an improved sample complexity analysis of asynchronous Q-learning, and a novel variance reduction algorithm and its analysis). We recommend the paper for acceptance and encourage the authors to account for the reviewers’ comments when preparing the camera-ready version of the paper.